# New Insight into the Molecular Pathomechanism and Immunomodulatory Treatments of Hidradenitis Suppurativa

**DOI:** 10.3390/ijms24098428

**Published:** 2023-05-08

**Authors:** Elisa Molinelli, Helena Gioacchini, Claudia Sapigni, Federico Diotallevi, Valerio Brisigotti, Giulio Rizzetto, Annamaria Offidani, Oriana Simonetti

**Affiliations:** Dermatological Unit, Department of Clinical and Molecular Sciences, Polytechnic Marche University, 60126 Ancona, Italy

**Keywords:** adalimumab, biologics, hidradenitis suppurativa, infliximab, secukinumab, therapy

## Abstract

Hidradenitis suppurativa (HS) is an immune-mediated inflammatory disorder characterized by deep-seated nodules, abscesses, sinus tracts and scars localized in the intertriginous areas. It is accompanied by pain, malodourous secretion and a dramatically decreased quality of life. Although the pathogenesis has not been entirely elucidated, the primary event is follicular hyperkeratosis of the pilosebaceous apocrine unit. Since the registration of the tumor necrosis factor-alpha inhibitor Adalimumab in 2015, several cytokines have been implicated in the pathomechanism of HS and the research of novel therapeutic targets has been intensified. We provide an update on the inflammatory cytokines with a central role in HS pathogenesis and the most promising target molecules of future HS management.

## 1. Introduction

Hidradenitis suppurativa (HS) is a chronic, inflammatory, recurrent, debilitating, cutaneous disease characterized by painful, deep-seated, inflamed lesions in the apocrine gland-bearing regions of the body (*Dassauer* definition) [1,2]. The prevalence of HS ranges from 1% to 4% in Europe and the United States in the general population, although the true incidence is probably underestimated. It usually manifests after puberty in the second and third decades of life, with a female predominance (female-to-male ratio of 3:1). Prepubertal onset (before 11 years of age) is estimated to occur in 2–7% of patients with HS [3,4,5].

The diagnosis of HS is clinical, and it is based on typical HS lesions with a predilection for intertriginous sites and recurrence. Disease severity is usually assessed through the Hurley score, the International Hidradenitis Suppurativa 4 (IHS4) score, the Dermatology Life Quality Index (DLQI) and the PAIN visual analogue scale [6]. Recently, high-frequency ultrasonography has been introduced in the clinical practice to confirm HS diagnoses, assess the disease severity and evaluate the treatment response [7]. HS could be staged sonographically using the classification proposed by Wortsman et al., the sonographic scoring system for HS (SOS-HS) that combined the results of the parameters included in Hurley’s classification with the relevant sonographic findings such as the detection of the widening of hair follicles, alterations in the dermal thickness or echogenicity, lesional types of involvement (pseudocyst, fluid collection, fistula) and lesion distribution (skin layer) and extension (cm) [8].

In addition, the color-doppler ultrasound could be used to better assess the presence and the morphology of sinus tracts and it is considered a valid imaging biomarker to evaluate responses during biologic treatments [9,10]. In the last decade, several diseases have been reported in association with HS, such as inflammatory bowel disease, hormonal disorders (polycystic ovarian syndrome), psoriasis and arthritis. Higher rates of comorbidities among patients with HS, including dyslipidemia, obesity, hypertension, diabetes and metabolic syndrome have been observed [11].

Treatment is a challenge because of the paucity of effective therapies and frequent exacerbations, with a negative impact on quality of life [12,13]. Medical approaches for HS currently consist of several treatment options, including topical and systemic approaches [14]. Topical treatments mainly consist of antiseptic washes, topical antibiotics and resorcinol 15% cream [15,16]. Systemic therapy includes primarily oral and intravenous antibiotics as monotherapy (tetracyclines, ertapenem, dalbavancin) or in combination (clindamycin and rifampicin) [17,18]. Retinoids, dapsone, oral zinc, contraceptive agents and immunomodulators are valid alternatives in the management of the disease alone or in combination [19,20]. Using lasers for hair removal (long-pulsed neodymium-doped yttrium aluminum garnet laser, alexandrite laser) considering the pathogenic role of hair follicles as a foreign body that stimulates the chronic inflammation in sinus tracts has emerged as a valid approach in patients with HS, as it reduces acute flares and prolongs the disease-free state [21,22,23]. To date, Adalimumab is the first and only FDA and EMA licensed biologic agent for the management of moderate-to-severe HS in adolescents and adults [24,25]. Immunomodulation targeting IL-1, IL-12/Th1 and IL-23/Th17 pathways is rapidly becoming the cornerstone of therapy for moderate-to-severe HS [26]. Surgical management represents the definitive treatment option, especially in severe and advanced HS. Surgical procedures range from incision and drainage for acute flares, de-roofing, narrow margin excision and wide local excision [14,27]. Wide local excision consists of the resection of the entire affected area with a wide lateral margin (1–2 cm). It is recommended by all guidelines as a surgical intervention to treat advanced regional diseases [27]. Several reconstructive techniques after surgical excision have been described, including primary closure, secondary intention healing, split-thickness skin grafts and skin flaps. Skin flaps are usually considered the best reconstructive approach. However, they may determine the displacement of potentially affected tissue and may lead to HS recurrence. Split-thickness skin grafts may cause inelastic contraction scars [14]. Recently, a new reconstructive surgical method using a cograft of an acellular dermal matrix (a full-thickness section of skin from a donor source such as a human cadaver, porcine or bovine in origin) and a split-thickness skin graft has been described as a promising wound-closure method after wide excisions, with rapid and good aesthetic results limiting scar formation [28]. To date, few studies on the best surgical approach in HS have been conducted, and there is still no consensus regarding the optimal surgical strategy and reconstructive techniques in HS [14,27]. Surgical treatments determine a significant improvement in the health-related quality of life of HS patients and result in a disease-free state and reduce the risk of recurrence [29]. However, the promising role of the combination of surgery and biologics that has synergistic effects needs to be further outlined [30].

## 2. Pathogenesis of HS

The exact pathogenesis of HS is not completely understood. It is assumed that the disease is triggered by genetic and environmental factors including smoking, obesity and skin occlusions. It has been speculated that the dysregulation of the gamma-secretase/Notch pathway is implicated in the basis of follicular occlusion in HS. Deficiency in the Notch signaling pathway, which has a pivotal role in maintaining the inner and outer root sheath of the hair follicle and skin appendages, results in the conversion of hair follicles to keratin-enriched epidermal cysts, compromises the apocrine gland homoeostasis and leads to the stimulation of toll-like receptor (TLR)-mediated innate immunity, which supports and maintains chronic inflammation [31,32,33]. The immunological features of HS mainly include the release of pathogen-associated molecular patterns (PAMPs) or damage-associated molecular patterns (DAMPs), the activation of T helper (Th)1 (TNFα) and Th17 (IL-23 and IL-17) pathways, the activation of macrophages mediated by the tool-like receptor (TLR) with the release of TNFα and the activation of the inflammasome with the increase in IL-1β production [34,35]. Specifically, aberrant TLRs signaling on macrophages and dendritic cells (DCs) induce an increased release of proinflammatory cytokines and chemokines as tumor necrosis factor (TNF)-alpha (α), interferon (IFN)-α interleukin (IL)-1β, IL-6 and IL-8, which stimulates the activation of dendritic cells. These activated DCs promote the activation of the adaptive immune system by inducing T-cell proliferation with the expansion of the Th1 and Th17 cell subsets, which contribute to increasing the release of proinflammatory cytokines including IL-17, IL-23 and TNFα. This in turn creates a positive feedback loop in which TNFα is an integral cytokine that propagates the chronic inflammatory cycle in the context of HS [36,37]. The etiopathogenesis of HS is multifactorial with genetic predispositions, environmental factors (smoking and obesity), hormonal factors and immune dysfunction being involved in both the onset and maintenance of the disease [38]. Recently, it has hypothesized that the interaction of endogenous and exogenous factors causes the activation of predominantly the innate immune system, which leads to perifollicular inflammation [39]. This causes hyperkeratosis and hyperplasia of the follicular epithelium, especially the infundibulum, which results in a follicular occlusion [40]. The dilatation and the rupture of the hair follicle results in the induction of an intense inflammatory immune response with the recruitment of neutrophils, macrophages, B-cells, Th1 and Th17 cells into the skin, which leads to inflammatory nodules or the formation of abscesses [38]. The immune cells produce several proinflammatory cytokines including Interleukin (IL)-1β, IL-6, CXC chemokine ligand (CXCL)/IL-8, IL-12p70, IL-23p40, IL-17A and IL-36 with a strong Th17 signature [38,41,42,43,44]. These proinflammatory cytokines are extensively considered key players in the immune dysregulation of the acute and chronic state of HS [35]. Several studies demonstrated that T cells and dendritic cells are responsible for the secretion of IL-23 and IL-12, which leads to a Th17 predominant immune response and keratinocyte hyperplasia [45,46]. IL-23 has been shown to induce IL-17-producing T helper cells, which infiltrate the dermis in HS lesions [47]. The IL-17 family has been shown to be important in the pathogenesis of several autoimmune and autoinflammatory diseases, especially psoriasis. IL-17 also plays an essential role in host defense against extracellular bacteria and fungi, and it has been shown to increase the expression of skin antimicrobial peptides/alarmins, such as hbD2 and psoriasin [46]. Blocking IL-17 therefore seems to be a valid therapeutic approach for HS as well [46]. During disease progression, several cytokines are found to be increased in HS lesional and perilesional skin. TNF-α has been shown to be elevated. These findings resulted in the introduction of anti-TNF-α agents in the therapy of HS. Along with HS progression, increased levels of TNF, IL-1β, IL-17, caspase-1 and IL-10 can be found in the tissue, which thus leads to the recruitment of neutrophils, mast cells and monocytes, which differentiate into macrophages and dendritic cells [47,48,49,50,51]. Recent evidence further points to an autoinflammatory mechanism in HS. HS skin shows an increased formation of neutrophil extracellular traps (NET). Immune reactions to neutrophil and NET-related antigens have been associated with increased immune deregulation and inflammation [52]. In addition to the role that the strong type I interferon (IFN) plays in HS skin, these findings indicate a pivotal role of the innate immune system in the pathogenesis of HS [53]. As healing from the inflammatory process moves on, tissue scarring progresses [45,46,47,48]. The development of scarring and sinus tracts is associated with metalloproteinase-2, tumor growth factor (TGF)-β and ICAM-1, with a possible augmentation of TGF-β and ICAM-1 signaling via specific components of the microbiome [45,46].

The role of bacteria in HS is still controversial. The principal bacterial species isolated in HS lesions include Gram-positive cocci including *Staphylococus aureus* and streptococcal species, coagulase-negative staphylococci (CoNs), Proteus mirabilis and mixed anaerobic bacteria [49]. *Staphylococcus epidermidis* and *Corynebacterium* spp. as well as atypical bacteria such as *Proteus mirabilis*, *Escherichia coli*, *Enterobacter aerogens* and *Enterococcus faecalis* were detected in microbiological samples of draining lymph nodes in HS patients after surgical excisions [50]. The bacterial superinfection of established lesions may contribute to maintaining chronic inflammation [49,51].

Recently, it has been hypothesized that periodontitis, which has been causally connected with the development of other immune-mediated inflammatory disorders such as psoriasis, may contribute to the development of HS. Significantly higher values of the average copy-count numbers of the total bacteria were found in HS patients. The majority of periodontal pathogens, especially *Treponema denticola*, were more frequently isolated in patients with HS than among the controls [52,53]. The use of antibiotics, which represent the first-line treatment of HS, is straightly related to their anti-inflammatory properties and immunomodulatory effects on T cells [54]. In addition, the worldwide increasing occurrence of antibiotic resistance suggests that the use of antibiotic treatment in HS should be specific and targeted based on microbiological evaluations [55]. A common mechanism of antibiotic resistance is represented by the production of biofilm [56]. It has been demonstrated that 89% of *S. epidermidis* isolated in patients with active HS lesions are strong bio-film producers in vivo [57]. Interestingly, a study on the role of Panton–Valentine leukocidin (PVL), which appears to be a virulence factor that exacerbates the pathogenicity of *Staphylococcus aureus* infections and especially induces severe necrotic, deep-seated skin infections, abscesses and recurrences in HS patients, was evaluated. The results seem to exclude the pathogenetic involvement of *Staphylococcus aureus* producing PVL in HS [58].

An increasing amount of literature has focused on the pathophysiological aspect of HS, and several molecules have been identified in histologic samples and classified as new potential therapeutic targets of the disease [59,60,61,62,63,64]. We summarized the literature data on the role that cytokines play in HS pathomechanism in order to add relevant information to our current understanding of HS. A brief mention on the preclinical and clinical research on the most promising therapeutic targets in HS was also included. We summarize the latest drugs and target interleukins (Figure 1).

## 3. TNF-α Generality and Role in HS Pathogenesis

TNFα is a member of the TNF superfamily, which includes at least 19 cytokines that exhibit a proinflammatory activity. TNFα is secreted as a 17-kDa soluble cytokine (sTNF) by innate and adaptive immune cells and nonimmune cells including macrophages, mast cells, granulocytes, B and T lymphocytes, natural killer cells, fibroblasts, neurons, smooth muscle cells and keratinocytes [65,66]. TNFα interacts with two transmembrane glycoprotein receptors: TNF receptor 1 (TNFR1 or p55), which is constitutively expressed in all cell types except erythrocytes, and TNF receptor 2 (TNFR2 or p75), which is predominantly expressed in endothelial and hematopoietic cells [67,68,69]. TNFα induces several signaling pathways leading to the activation of nuclear factor kappa-B1 (NF-κB1), caspase-8- and caspase-3-dependent apoptosis, extracellular signal-regulated kinases (ERK), p38 mitogen-activated protein kinases (p38MAPK) and c-Jun N-terminal kinases (JNK) [70]. The proinflammatory effect of TNFα is mediated through the NF-kB1 pathway, which promotes the rapid induction of inflammatory cytokines (IL-1β, IL-2, IL-6, IL-8) and adhesion molecules (intercellular adhesion molecule-1, vascular adhesion molecule-1), especially in monocytes and tissue macrophages [71].

In 2001, Martinez et al. firstly described the relationship between the TNF-α blockade and HS and observed an improvement in HS manifestations in a patient with Crohn’s disease treated with a TNF-α inhibitor (infliximab) [72]. Successively, several studies demonstrated an increased TNF-α expression in HS skin lesions [73]. Emelianov et al. reported significantly higher levels of TNFα in the epidermis and dermis of HS skin compared with that of the control, except for the immunoreactivity of TNFα at the proximal outer root sheath, which was decreased compared with the control skin [74]. Matusiak et al. described significantly high serum levels of TNFα in HS patients compared with healthy controls; the level of cytokine did not correlate with the disease severity, its duration or body mass index [75]. Van der Zee et al. showed an increased expression of proinflammatory TNF-α and IL-1β as well as anti-inflammatory Il-10 in patients with HS compared with the healthy controls in both lesional and perilesional skin. It was also five-times higher than the values observed in those with psoriasis. Moreover, an elevated expression of TNF receptors (TNF-R1 and TNF-R2) was detected in HS lesions [43,76]. Similar conclusions regarding the level of TNF-α in patients with HS have been reached by Mozeika et al., who reported elevated levels of cytokines in the skin, apocrine glands and hair follicles of HS patients [77]. The role of TNF-α in the HS is multifaceted. Firstly, TNF-α seems to be implicated in the relationship between smoking and HS. Nicotine acetylcholine receptors are expressed in keratinocytes, sebocytes, mast cells, neutrophils, lymphocytes and macrophages, which are all implicated in the HS pathogenesis. It is known that nicotine causes hyperkeratosis and hyperplasia of the follicular epithelium, occlusion of the follicular ostia and rupture of the enlarged hair follicle [78,79]. Nicotine stimulates eccrine gland secretion and TNF-α release by keratinocytes and Th17 cells. Nicotine directly induces macrophages to release IL1β, TNF-α and matrix metalloproteases (MMPs) [80]. Moreover, TNF-α mediates the release of MMP2 and MMP9, which have a crucial role in the excessive inflammatory response and tissue injury seen in the lesional skin of HS [80]. In addition, Moran et al. demonstrated that treatments with TNF inhibitors induce a significant decrease in IL-17-expressing CD4 T cells in HS skin [81].

Thus, TNF-α increments the ratio of Th17 to regulatory T-cells, which leads to a Th17 predominant immune response and keratinocyte hyperplasia [81]. Finally, TNF-α seems to be implicated in the correlation between HS and metabolic comorbidities, particularly in terms of higher fasting serum-glucose insulin levels and insulin resistance. It has been demonstrated that adiponectin levels are considerably decreased in HS patients [82]. TNF-α inhibits the release of adiponectin, which is an anti-inflammatory adipokine that regulates glucose metabolism and insulin sensitivity. It is negatively correlated with body mass index (BMI), and decreased circulating levels are associated with diabetes and metabolic syndrome [83,84].

### 3.1. Adalimumab

Adalimumab is a fully humanized monoclonal antibody against soluble and transmembrane TNFα. It is the most well-studied biologic agent in HS with multiple case reports, case series and randomized controlled trials conducted. Indeed, it is the only biologic agent approved for the treatment of moderate-to-severe HS in adult and adolescent patients by the FDA and EMA [85]. Its efficacy was tested in two large, double-blind, placebo-controlled studies (PIONEER I and PIONEER II) with 633 patients enrolled and a Hidradenitis Suppurativa Clinical Response (HiSCR) was reached by 50% of patients versus 26% in the placebo group. The long-term efficacy of adalimumab was demonstrated in an open-label extension of these two studies that confirmed similar results in 151 patients followed longitudinally for at least 96 weeks (an HiSCR was achieved in 52% of patients) [86,87]. The real-life efficacy of adalimumab was confirmed in a retrospective, multicenter cohort study including 389 patients with HS in which its safety and efficacy were consistent with randomized controlled trials (an HiSCR was achieved in 54% of patients) [25] (Table 1).

### 3.2. Infliximab

Infliximab (IFX) is a chimeric monoclonal antibody whose target is soluble and transmembrane TNF-α [88]. In a phase II double-blind study on infliximab, 38 patients with moderate-to-severe HS were randomized to receive infliximab 5 mg/kg on weeks 0, 2, 4, 6, 14 and 22 or a placebo. The results showed that 4/15 (27%) of the infliximab patients vs. 1/18 (5%) of the placebo patients achieved the primary endpoint of a ≥50% decrease in the Hidradenitis Suppurativa Severity Index (HSSI) score [89]. In a long-term study on 10 patients, 3 patients reported no relapse at a 2-year follow-up [90]. Recently, the real-life efficacy of infliximab was corroborated in a retrospective cohort study of 52 patients with HS: 67% of the patients achieved the primary endpoint of HS stability by receiving 10 mg/kg administered every 6–8 weeks [91]. Van Rappard et al. retrospectively compared two cohorts of 10 patients with severe, recalcitrant HS, whereby one group was treated with intravenous 5 mg/kg infliximab at weeks 0, 2 and 6, the other one was treated with adalimumab 40 mg every other week. The authors showed that the HS severity (measured with the Sartorius score and DLQI) diminished in both groups but adalimumab was less effective than infliximab probably due to a lower dosage and absence of an induction phase of Adalimumab [92] (Table 1).

### 3.3. Certolizumab Pegol

Certolizumab pegol is a pegylated fragment antigen binding (Fab) region of a humanized antibody that binds to TNF-α [93]. A retrospective review reported the inefficacy of certolizumab in two patients with HS, who previously failed to respond to other TNFα inhibitors. The successful use of certolizumab pegol in patients with moderate-to-severe HS refractory when taking other biologic agents has been reported in several case reports [94,95,96,97]. In one case report, the authors described the use of certolizumab in a pregnant patient presenting both HS and psoriasis [97] (Table 1).

### 3.4. Etanercept

Etanercept is a recombinant human TNF inhibitor that acts as a soluble TNF receptor and binds TNF-alpha and TNF-beta [98].

Several open-label studies reported the efficacy and safety of etanercept in the management of HS [99,100,101,102]. In a randomized double-blind trial assessing the efficacy of etanercept in 20 HS patients, no significant difference was found in PGA and DLQI between the Etanercept and placebo cohorts [103] (Table 1).

### 3.5. Golimumab

Golimumab is a fully human anti-TNF-α monoclonal antibody that targets both soluble and transmembrane TNF-α [104]. Three case reports describing the use of golimumab in four HS patients are present in the literature. In one patient, 50 mg of golimumab monthly did not result in a clinical improvement in HS [105]. In the other three patients, a higher dosage of golimumab (200 mg followed by 100 mg monthly) showed brilliant results with a decrease in IHS4 [106,107]. Golimumab could be an alternative therapy in HS, especially in those with HS-associated inflammatory arthritis and adalimumab failure, as patients show improvement in both diseases [107] (Table 1).

### 3.6. IL-17 Generality and Role in HS Pathogenesis

IL-17A, commonly referred to as IL-17, is a member of the IL-17 cytokine family that includes six members alongside IL-17F. IL-17 is a proinflammatory cytokine that binds the IL-17 receptor (IL-17RA), which is expressed in different immune and nonimmune cells, such as endothelial cells, fibroblasts, osteoblasts, keratinocytes, monocytes and macrophages. This linkage results in the activation of transcription nuclear factor-KB, which implies proinflammatory gene expression [108].

IL-17 induces the release of several chemokines with the majority being chemokine CXC ligand 1, chemokine CXC ligand 8 and chemokine CC ligand 20, which mediate the recruitment of neutrophils, macrophages and lymphocytes; stimulate both the adaptive and innate immune systems; and sustain tissue inflammation. T helper 17 (Th17) cells, the main producers of IL-17, express chemokine receptor 6 themselves, which is the receptor for chemokine CC ligand 20, and thereby they facilitate homing to the inflammatory sites and amplifying the immune response [108]. Moreover, IL-17 stimulates through a mechanism involving the nod-like receptor protein 3 (NLRP3) inflammasome and the production in macrophages of transforming growth factor-β (TGF-β), IL-1β, IL-6 and TNF-α [43,109,110,111,112,113,114,115], which are crucial for Th17 differentiation, proliferation and a subsequent increase in IL-17 production. TGF-β induces RAR-related orphan receptor gamma-γt, a transcription factor that is necessary for the generation of the Th17 cell lineage. IL-6 downregulates Foxp3, a transcription factor, which generates regulatory T cells [109,114,116]. IL-1β is produced by monocytes and macrophages and downregulates Foxp3 independently of IL-6 [116]. The presence of increased levels of Th17 cells in peripheral circulation and HS lesional skin, compared with healthy control subjects, was demonstrated in isolated studies. In 2010, Schlapbach et al. [112] reported a 30-fold increase in IL-17A gene expression in lesional HS skin compared with normal skin via a real-time polymerase chain reaction analysis. The authors also demonstrated that IL-17 is an inducer of beta-defensin-2, an antimicrobial peptide overexpressed in HS lesions. Similarly, Kelly et al. [37] observed a 149-fold increase in IL-17 mRNA expression in lesional HS skin versus normal skinAsymptomatic perilesional skin had a 50-fold increase in interleukin-17 gene expression; the presence of such large amounts of interleukin-17 mRNA in perilesional skin suggests a pathogenic function because inflammatory responses, although subclinical, were seen prior to the formation of active lesions. 

In addition, Matusiak et al. demonstrated elevated serum TNF-α and IL-17 levels in two studies [75,117]. According to them, TNF-α levels were not associated with disease severity, while IL-17 levels were higher in more severe diseases. In contrast, Blok et al. compared the serum sIL-2R, TNF-α, IL-17A and IL-17F levels of 12 patients with both healthy controls and after ustekinumab treatment and did not find any significant differences [115,118]. Interestingly, the IL-23/IL-17 axis seems to be involved in follicular hyperkeratinization, the increased expression of adenoside monophosphates and the accumulation of neutrophils in active acne, which is a disease that is highly linked to HS [119]. These observations provide the rationale for a new generation of biologics for HS that target IL-17A (secukinumab, ixekizumab) or its receptor (brodalumab) (Table 1).

### 3.7. Secukinumab

Secukinumab, a monoclonal antibody specifically targeting interleukin-17A, was the first anti-interleukin-17A drug to be approved for the treatment of psoriasis by the Food and Drug Administration and the European Medicines Agency. More recently, the Food and Drug Administration has approved ixekizumab, another anti-interleukin-17A drug, and brodalumab, an interleukin-17 receptor antagonist. Investigational trials of IL-17 therapy in HS are ongoing [119]. Since 2017, several case reports on the successful use of secukinumab in HS were described in the literature [120,121,122,123,124,125]. In two open-label trials of a total of 29 patients with moderate-to-severe HS, 300 mg of secukinumab was administered subcutaneously once a week for 5 weeks and then every 4 weeks or every 2 weeks for 24 weeks. An HiSCR was achieved in 20/29 patients (68%) [126,127]. Comparable results of the same protocol were observed in two retrospective studies assessing 37 HS patients, whereby an HiSCR was met in 22 patients (59%) [128,129]. Most commonly, adverse events included gastrointestinal upset such as Crohn’s disease, so caution is necessary for HS patients with a higher risk for inflammatory bowel disease [128]. Currently, there are three phase III randomized clinical trials of secukinumab for the treatment of HS (NCT03713619, NCT03713632, NCT04179175) [130,131,132] (Table 1).

### 3.8. Brodalumab

Brodalumab is a fully human IgG2 monoclonal antibody that binds the IL-17RA subunit of the IL-17 receptor and thus interferes with the signaling of various isoforms of IL-17 (mostly IL-17A, IL-17C and IL-17F). The FDA approved brodalumab for moderate-to-severe plaque psoriasis [133]. Three case reports have been reported on the promising use of brodalumab in HS patients [134,135,136]. Brodalumab was evaluated in an open-label cohort study of 10 patients with moderate-to-severe HS at the dosage of 210 mg sc at weeks 0, 1 and 2 and then every 2 weeks. At week 12, all patients achieved an HiSCR. Some of them had an extremely rapid clinical response (2 weeks) that mostly involved a decrease in tunnel drainage [137]. A different protocol provided brodalumab weekly after the loading dose was assessed in an open-label cohort study of 10 patients. All ten patients reached an HiSCR at week 4, 80% achieved an HiSCR at week 7 and 50% achieved an HiSCR at week 12 [138]. No major adverse events were experienced. The last protocol is currently being tested in a new phase I clinical trial (NCT04979520) [139] (Table 1).

### 3.9. Ixekizumab

Ixekizumab is a humanized IgG4 monoclonal antibody that binds soluble IL-17A and IL-17 A/F. It is FDA approved in moderate-to-severe plaque psoriasis, psoriatic arthritis, ankylosing spondylitis and nonradiographic axial spondyloarthritis [140]. In the literature, there are two case reports of HS and concomitant psoriasis effectively treated with ixekizumab and one case report of HS alone also being successfully managed with this biologic agent [141,142,143]. Esme P et al. [144] recently described five cases of Hurley stage III HS patients treated with ixekizumab. Four of the five patients (80%) achieved an HiSCR, and no adverse event was recorded (Table 1).

### 3.10. Bimekizumab

Bimekizumab is a humanized IgG1κ monoclonal antibody that selectively targets IL-17A and IL-17F and is used in patients with plaque psoriasis [145]. The efficacy of bimekizumab was tested in a phase II clinical trial including 90 patients with moderate-to-severe HS, in which bimekizumab was compared with a placebo and Adalimumab. At week 12, 46% of patients in the bimekizumab branch reached an HiSCR, and 32% of them achieved a near HiSCR. These results were superior compared to those of the placebo and Adalimumab branch [146]. The use of bimekizumab in HS is currently being evaluated in three phase III clinical studies that are currently ongoing (NCT04242446, NCT04242498, NCT04901195) [147,148,149] (Table 1).

### 3.11. IL 23

As mentioned above, there is increasing evidence of the significant role of the IL-23/IL-17 cytokine axis in the pathogenesis of HS. IL-23 is a heterodimeric cytokine containing the identical p40 subunit linked to the p19 subunit. The main sources of IL-23 in HS are monocytes/macrophages and dendritic cells. IL-23 plays a central role in Th17 lineage activation as it promotes the survival and proliferation of Th17 and stimulates the release of IL-17, IL-22, IL 1-β and TNFα [150]. IL-12 is a cytokine that is structurally related to IL-23, which is involved in the upregulation of T-cell immune responses. IL-12 is a heterodimeric protein consisting of two disulfide-linked glycosylated p35 and p40 subunits. The cytokine is synthetized by dendritic antigen-presenting cells (DCs) and macrophages. It regulates the inflammatory cascade and promotes NK cell activation, T-cell differentiation and the expansion of the Th1 phenotype, which produces TNF, IL-2 and interferon (IFN)-γ [150,151,152]. The p40 subunit of both IL-12 and IL-23 binds to the IL-12 receptor-β1 (IL-12R β1) while the IL-12 p35 and IL-23p19 subunit bind to IL-12R β2 and IL-23R, respectively. Thus, despite the structural similarities, IL-12 and IL-23 regulate distinct inflammatory cytokine pathways [153,154]. IL-12 contributes to Th1 proliferation by stimulating IFN-γ, IL-2 and TNFα synthesis. Conversely, IL-23, in association with IL-21, promotes the activation of Th17 cells and the production of several proinflammatory mediators such as IL-17 and TNFα [155]. IL-23 plays a role in the final maturation of Th17 cells by producing a downstream impact on Th1 and Th17 cell activation and stimulating the secretion of proinflammatory cytokines and angiogenic factors. Additionally, IL-23, which promotes the survival and proliferation of Th17 cells, is a potent activator of keratinocyte proliferation, which enables the further stimulation of keratinocyte proliferation and the secretion of additional proinflammatory mediators in HS [156]. The hypothesis that IL-23 may have a key role in the pathogenesis of HS was based on the detection of increased levels of IL-17 and IL-23 in HS lesional and nonlesional HS skin compared to healthy controls. Dajnoki et al. demonstrated that the dermal production of IL-23 and TNF-a was significantly enhanced only in the lesional HS skin together with significantly increased T cell, dendritic cell and macrophage influx and elevated IL-12, IFN-c, IL-17A, IL-10, TGF-b and CCL20 expression levels. In lesional skin, epidermal IL-1b, IL-23, TNF-a and CCL2 protein levels also remained high compared with nonlesional HS [156] (Table 1).

### 3.12. Ustekinumab

The anti-p40 drug, ustekinumab, which binds to the p40 subunit common to interleukin-12 and interleukin-23, has also shown considerable improvements in patients with psoriasis, psoriatic arthritis and Crohn’s disease and is currently used for the treatment of these diseases [157]. Similarly, ustekinumab has also shown efficacy in treatment of HS, as described in two case series [158,159]. In a phase II open-label study of 17 patients with moderate-to-severe HS treated with ustekinumab at a psoriasis dosage, 47% of patients achieved an HiSCR [115]. A different protocol was evaluated in a multicentric retrospective review of 14 HS patients treated with ustekinumab at dosage regimens for Crohn’s disease (consisting of intravenous induction): 50% of patients achieved an HiSCR and 71% had a significant pain reduction and improvement in their quality of life [160]. Similar results were achieved in a prospective study of six HS patients that were also treated with Crohn’s dosage protocol [161] (Table 1).

### 3.13. Guselkumab

The IL-23 pathway is also involved in HS pathogenesis, so its blockade could contribute to reaching disease control. Guselkumab is a monoclonal antibody that inhibits the p19 subunit of extracellular IL-23, and it is currently approved for psoriasis in adults [162]. Recently, authors have reported its effectiveness in several case reports and case series in patients with a moderate-to-severe HS refractory to other systemic treatments [163,164,165,166,167]. However, adequate dosing and intervals have not been determined yet, so in most published series, doses approved for psoriasis are commonly used (100 mg at weeks 0 and 4 and then every 8 weeks) [168]. Less encouraging results were obtained in another case series in which 100 mg of guselkumab was administered with an increased frequency (week 0, and then every 4 weeks) [169]. A phase II placebo-controlled, double-blind study (NCT03628924), reported that an HiSCR was reached in 50% of the participants treated with 200 mg of guselkumab at weeks 0, 4, 8 and 12. In comparison, an HiSCR was achieved in 38% of the participants in the placebo group and in 45% of patients receiving 1200 mg intravenously at weeks 0, 4 and 8, followed by 200 mg of guselkumab subcutaneously starting from week 12 [170] (Table 1).

### 3.14. Risankizumab

Risankizumab is a fully human IgG1κ monoclonal antibody that selectively blocks IL-23 by binding the p19 subunit, and it is used to treat psoriasis and/or psoriatic arthritis [171]. Risankizumab was successfully administered in four HS patients in three distinct case reports [172,173,174]. Recently, Repetto et al. [175] described a case series of six patients with HS treated with risankizumab. Three patients reached an HiSCR at month 3, and all patients achieved an HiSCR at month 6. Currently, a phase II placebo-controlled study evaluating patients with moderate-to-severe HS randomized to receive one of two dose levels of risankizumab or a placebo is ongoing (NCT03926169) [176] (Table 1).

### 3.15. Tildrakizumab

Tildrakizumab is a humanized IgG1κ monoclonal antibody targeting the p19 subunit of IL-23, and it is approved for use to treat moderate-to-severe plaque psoriasis [177]. In the literature, only two case series describe the administration of 100 mg of tildrakizumab subcutaneously at weeks 0 and 4, and then 200 mg monthly in patients with moderate-to-severe HS. All patients achieved an HiSCR [178,179] (Table 1).

### 3.16. IL1

The IL-1 family is composed of 11 cytokine members, of which seven act as agonists (IL-1α, IL-1β, IL-18, IL-33, IL-36α, IL-36β and IL-36γ) and four as antagonists (IL-1 receptor antagonist (Ra), IL-36Ra, IL-37 and IL-38) [180]. IL-1α and IL-β are proinflammatory cytokine targets in inflammatory diseases. IL-1α is normally expressed in hematopoietic immune cells and as well as in other cell types including keratinocytes [181]. Although IL-1α is usually located in the membranes, it is also expressed intracellularly in the cytosol and in the nucleus, whereby it activates the transcription of genes that promote inflammation, tissue homeostasis and repair [182]. Proinflammatory stimuli induce IL-1α expression, which lead to IL-1R1 binding and proinflammatory gene expression involving type 1 or type 17 immune responses. The enrollment and activation of T cells, dendritic cells, neutrophils and macrophages provoke the release of further proinflammatory cytokines and chemokines, which thus lead to an autoinflammatory amplification loop [183]. IL-1β is a circulating cytokine, and its expression is inducible only in monocytes, macrophages and dendritic cells [183]. Full-length IL-1β precursor protein (pro-IL-1β), normally present in the cytoplasm, is cleaved by caspase-1 into its active form in response to the activation of pattern recognition receptors (PPR) by PAMPs or dDAMPs in an inflammasome-dependent process [184]. Moreover, pro-IL-1β can also be activated in the extracellular space by neutrophils and mast cell-derived proteases or by microbial proteases [185]. The antagonist IL-1Ra binds to the IL-1R1 receptor and competes with IL-1α and IL-β and thus exerts an anti-inflammatory response [186].

The IL-1 family is considered a key mediator of the innate immune system as it maintains endogenous hemostasis and links innate and adaptive responses. Hyperactivation of the IL-1 pathways seems to play a central role in the pathogenesis of HS. IL-1β released by dendritic cells and macrophages/monocytes induces a wide production of chemokines, which leads to neutrophile infiltration and inflammation. IL-1β also increases the secretion of MMPs, which results in tissue destruction [187].

In 2021, Witte-Händel et al. demonstrated that IL-1β was strongly expressed in lesional HS skin compared with healthy skin, in contrast with IL-1RA levels; accordingly, this leads to an increased IL-1 β/IL-Ra ratio. The overexpression of these cytokines is more conspicuous in HS skin than in lesional psoriatic skin [187].

Recently, Wolk at al. reported that the expression of granulocyte colony-stimulating factor (G-CSF), a key regulator of neutrophil survival and function, resulted in an incremented in both the skin lesions and blood of HS patients. Moreover, blood levels were positively correlated with disease severity [188]. In cellular models, IL-1β, IL-17 and IL-36 induce the expression of G-CSF in fibroblasts and dendritic cells [188,189].

Finally, IL-1α is a potent inducer of the production of VEGF. Angiogenesis may play a role in the pathogenesis of HS. Conversely to psoriasis, HS keratinocytes exhibited a significant lower level of VEGF, as well as IL-1α and IL-22 compared to normal keratinocytes using an in vitro scratch assay, which suggests that changes in VEGF signaling may be associated with HS pathogenesis [190,191] (Table 1).

### 3.17. Anakinra

Interestingly, according to a small randomized clinical trial, anakinra, an IL-1 receptor antagonist, has the potential to be an effective and safe treatment for severe HS [192]. Additionally, the beneficial effect of anakinra was prolonged [193]. Both comparable and conflicting results have been published for anakinra treatment in HS in one small open study and in several case reports. Anakinra is a fully human recombinant monoclonal antibody that impedes the interaction between IL-1α and IL-1β and their receptor IL-1R. It is FDA approved for use in rheumatoid arthritis and neonatal-onset multisystem inflammatory disease [192].

In a placebo-controlled trial, 20 HS patients were randomized to receive 100 mg of anakinra subcutaneously once daily or a placebo for 12 weeks. At week 12, 78% of patients in the anakinra group achieved an HiSCR, compared to 30% in the placebo group. These results were not maintained after the end of the treatment due to an HS relapse [193]. Similar results, including HS relapse after the discontinuation of anakinra, were obtained in a case series of six HS patients treated with the same protocol [194]. Conversely, disease relapse did not occur in a case report of an HS patient treated with a higher dose of anakinra (200 mg once daily) [195].

Finally, several case reports described the failure of anakinra in treating patients with severe disease [105,196,197] (Table 1).

### 3.18. Bermekimab

Bermekimab is a fully human recombinant IgG1κ monoclonal antibody that inhibits IL-1α [198]. In a randomized trial of 20 HS patients, the results showed that 75% of patients receiving 7.5 mg/kg of bermekimab intravenously every other week achieved an HiSCR [199,200]. A phase II open-label study reported that a mean of 62% of HS patients treated with 400 mg of bermekimab subcutaneously weekly achieved an HiSCR [201]. Currently, a phase II, randomized, placebo and active comparator-controlled dose range study evaluating bermekimab use in HS patients is ongoing (NCT04988308) [202] (Table 1).

### 3.19. Canakinumab

Canakinumab is a fully human IgG1κ monoclonal antibody that targets IL-1β [203]. There are controversial results concerning canakinumab use in HS patients. Two case reports describing three HS patients showed a positive response to canakinumab [204,205], while two other case reports describing a total of three patients reported canakinumab failure with this disease [206,207] (Table 1).

### 3.20. IL36

The IL-36 subfamily exerts a pivotal role in regulating the innate immune system. It is composed of three agonists with proinflammatory activity (IL-36α, IL-36β and IL-36γ) and three antagonists (IL-36RN or IL-36Ra, IL37 and IL-38) [208]. IL-36 is constitutively expressed in epithelial and immune cells; it is involved in proinflammatory signaling pathways and in the regulation of innate and adaptive immune responses [209].

Agonists bind to receptor complex IL-36R and promote the activation of NF-kB and mitogen-activated protein kinases, which leads to T-cell proliferation; Th1 lymphocyte secretion; and the production of proinflammatory cytokines, chemokines and costimulatory molecules by dendritic cells. This results in the overexpression of several proinflammatory cytokines including IL-1β, IL-12, IL-23, IL-6, TNF-α, CCL1, CXCL1, CXCL2, CXCL8 and GM-CSF [208]. IL-36α and IL-36γ are mostly produced by keratinocytes but also by dermal fibroblasts, endothelial cells, macrophages, Langerhans and dendritic cells. In contrast with other IL-1 family cytokines, IL-36 cytokines are also generated as precursors but do not contain a caspase cleavage site. After their secretion, they are activated by proteases produced by neutrophils and present at neutrophil extracellular traps (NETs) such as elastase, cathepsin G and proteinase 3, and by cathepsin S, which is released by keratinocytes and fibroblasts [210,211,212]. In addition, keratinocytes secrete the protease inhibitors alpha-1-antitrypsin and alpha-1-antichymotrypsin (encoded by SERPINA1 and SERPINA3 genes), which impede the processing of IL-36 cytokines by neutrophil proteases and thus controls the inflammatory loop [186,213].

Hessam et al. demonstrated that IL-36α, -β and -γ and IL-36RA expression was upregulated in HS lesions in comparison with healthy skin. IL-37 and IL-38 were significantly higher in perilesional HS skin compared with healthy skin but decreased in lesional skin. Recent studies have confirmed a higher expression of all three IL-36 isomers in HS lesions compared to healthy controls; however, in these studies, IL-36RA was not significantly expressed in the HS lesional skin [214,215,216]. The discrepancy of the findings of, to our knowledge, the first three studies of the IL-36 cytokine family in patients with HS may be due to several factors, most importantly the selection of the skin samples, the skin region and the variation in disease severity [214,215,217] (Table 1).

### 3.21. Spesolimab

Spesolimab is a humanized monoclonal antibody that blocks the IL-36 receptor (IL-36R) [26]. The literature on anti-IL-36 agents in inflammatory skin disorders is smaller. There are no published clinical trials aimed at the IL-36 cytokine complex in HS. Currently, there is only one randomized phase II trial (NCT04762277) evaluating spesolimab efficacy in HS patients [218]. We look forward to future studies that will determine the role of IL-36 in the pathogenesis of HS and could hopefully be a new basis for treatment development [217] (Table 1).

### 3.22. Imsidolimab (ANB019)

Imsidolimab is a humanized monoclonal antibody that inhibits the activation of IL-36R, and it is currently being investigated in a phase II trial (NCT04856930) [219] (Table 1).

**Table 1 ijms-24-08428-t001:** Randomized controlled trials (RCTs), cohort studies, case series, case reports on biologic therapy in HS.

Drug	Study	Patients and Dose	Results
**TNFα**			
**Adalimumab**			
	Kimball et al., 2016 [86]	**PIONEER I (n = 307)** Adalimumab 40 mg/week s.c. (n = 153) Placebo s.c. (n = 154) **PIONEER II (n = 326)** Adalimumab 40 mg/week s.c. (n = 163) Placebo s.c. (n = 163)	41.8% reached an HiSCR after 12 weeks 26.0% reached an HiSCR after 12 weeks 58.9% reached an HiSCR after 12 weeks 27.6% reached an HiSCR after 12 weeks
	Zouboulis et al., 2019 [87]	**Open-label extension (OLE) trial (n = 151)** Adalimumab 40 mg/week s.c.	An HiSCR was achieved in 52% of patients at week 168
	Marzano et al., 2021 [25]	**Retrospective, real-life multicenter cohort** **study (n = 389)** Adalimumab 160 mg s.c. at week 0, 80 mg at week 2 and 40 mg weekly starting at week 4	An HiSCR was achieved in 43.7% at week 16 and in 53.9% at week 52
**Infliximab**			
	Grant et al., 2010 [89]	**Phase II placebo-controlled, double-blind RCT (n = 38)** Infliximab 5 mg/kg i.v. at weeks 0, 2, 4, 6, 14 and 22 (n = 15) Placebo i.v. (n = 23)	>50% decrease in HSSI reached in 27% at week 8 >50% decrease in HSSI reached in 5% at week 8
	Mekkes et al., 2008 [90]	**Long-term efficacy study (n = 10)** Single course of infliximab (three intravenous infusions at weeks 0, 2 and 6)	3/10 reported no relapse at 2-year follow-up 7/20 showed recurrence after 8.5 months
	Oskardmay et al., 2019 [91]	**Retrospective cohort study (n = 52)** Infliximab 10 mg/kg i.v. every 6 or 8 weeks	67% of the patients achieved HS stability
	Van Rappard et al., 2012 [92]	**Retrospective comparative study (n = 20)** Infliximab i.v., 3 infusions of 5 mg/kg at weeks 0, 2 and 6 Adalimumab s.c. 40 mg every other week	Average Sartorius score was reduced to 54% of baseline Average Sartorius score was reduced to 66% of baseline
**Certolizumab pegol**			
	Sand et al., 2015 [94]	**Retrospective study (n = 2)** Certolizumab pegol 200 mg s.c. every 2 weeks	Inefficacy with low dosage certolizumab
	Holm et al., 2020 [95]; Esme et al., 2020 [95]; Wohlmuth-Wieser et al., 2020 [97]	**Case reports (n = 3)** Certolizumab pegol 400 mg every other week or 200 mg every week	Satisfactory response to certolizumab
**Etanercept**			
	Giamarellos-Bourboulis et al., 2008 [99]	**Prospective open-label phase II study (n = 10)** Etanercept 50 mg s.c. once weekly for 12 weeks	>50% decrease in disease activity in 7 patients at week 24
	Pelekanou et al., 2010 [100]	**Open-label phase II prospective trial (long term efficacy study) (n = 10)** Etanercept 50 mg s.c. once weekly for 12 weeks	3/10 did not report any disease recurrence 7/10 needed a second course of treatment, of which 5 had favorable response and 2 were not successfully treated
	Cusack et al., 2006 [101]	**Open-label study (n = 6)** Etanercept 25 mg s.c. twice weekly	Mean reduction of 61% in self-reported disease activity at 24 weeks Mean reduction of 64% in DLQI scores at 24 weeks
	Sotiriou et al., 2009 [102]	**Open label study (n = 4)** Etanercept 25 mg s.c. twice weekly	68.75% mean self-reported improvement at 6 months follow-up Mean reduction of 66.5% in DLQI scores at 6 months follow-up
	Adams et al., 2010 [103]	**Randomized double-blind trial (n = 20)** Etanercept s.c. 50 mg every other week (n = 10) Placebo s.c. (n = 10)	No statistically significant difference between etanercept and placebo groups in PGA and DLQI
**Golimumab**			
	van der Zee et al., 2013 [105];; Tursi et al., 2016 [106]; Ramos et al., 2022 [107]	**Case reports (n = 4)** Golimumab 50 mg s.c. every 4 weeks Golimumab 200 mg s.c. at week 0 and 100 mg every 4 weeks	No clinical improvement Successful results with decrease in IHS4
**IL-17**			
**Secukinumab**			
	Thorlacius et al., 2018 [120]; Schuch et al., 2018 [121]; Jørgensen et al., 2016 [122]; Głowaczewska et al., 2020 [123]; Villegas-Romero et al., 2020 [124]; Chiricozzi et al., 2020 [125]	**Case reports (n = 8)** Secukinumab 300 mg s.c. weekly for 5 weeks, then every 4 weeks	Successful response to secukinumab
	Prussick et al., 2019 [126]	**Open-label, single-arm, pilot trial (n = 9)** Secukinumab 300 mg s.c. weekly for 5 weeks, then every 4 weeks	An HiSCR was achieved in 67% at week 24
	Casseres et al., 2020 [127]	**Open-label, single-arm pilot trial (n = 20)** Secukinumab 300 mg s.c. weekly for 5 weeks, then for 9 patients 300 mg s.c. every 4 weeks, and for 11 patients 300 mg s.c. every 2 weeks	An HiSCR was achieved in 70% at week 24
	Reguiaï et al., 2020 [128]	**Retrospective study (n = 20)** Secukinumab 300 mg s.c. weekly for 5 weeks, then every 4 weeks	An HiSCR was achieved in 75% at week 16
	Ribero et al., 2021 [129]	**Multicentric retrospective study (n = 31)** Secukinumab 300 mg s.c. weekly for 5 weeks, then every 4 weeks	An HiSCR was achieved in 41% at week 28
	NCT03713619 [130], NCT03713632 [131], NCT04179175 [132]	**Phase III RCTs**	Still ongoing
**Brodalumab**			
	Tampouratzi et al., 2019 [134]; Yoshida et al., 2021 [135]; Arenbergerova et al., 2020 [136]	**Case reports (n = 3)** Brodalumab 210 mg s.c. at weeks 0, 1 and 2, then 210 mg every 2 weeks	Successful response to brodalumab
	Frew et al., 2020 [137]	**Open-label cohort study (n = 10)** Brodalumab 210 mg s.c. at weeks 0, 1 and 2, then 210 mg every 2 weeks	100% achieved an HiSCR at week 12
	Frew et al., 2021 [138]	**Open-label cohort study (n = 10)** Brodalumab 210 mg s.c. every 2 weeks	100% achieved an HiSCR at week 4
	NCT04979520 [139]	**Phase III RCTs**	Still ongoing
**Ixekizumab**			
	Odorici et al., 2020 [141]; Megna et al., 2020 [142]; Reardon et al., 2021 [143]	**Case reports (n = 3)** Ixekizumab 160 mg s.c. at week 0; 80 mg at weeks 2, 4, 6, 8, 10 and 12 and then every 4 weeks	Successful response to ixekizumab
	Esme et al., 2022 [144]	**Case series (n = 5)** Ixekizumab 160 mg s.c. at week 0; 80 mg at weeks 2, 4, 6, 8, 10 and 12	80% achieved an HiSCR at week 12
**Bimekizumab**			
	Glatt et al., 2021 [146]	**Phase II, double-blind, placebo-controlled randomized clinical trial (n = 90)** Bimekizumab 640 mg s.c. at week 0 and 320 mg every 2 weeks Placebo s.c. Adalimumab 160 mg at week 0, 80 mg at week 2 and 40 mg every week after	57.3% achieved an HiSCR at week 12 26.1% achieved an HiSCR at week 12 60% achieved an HiSCR at week 12
	NCT04242446 [147]; NCT04242498 [148]; NCT04901195 [149]	**Phase III RCTs**	Still ongoing
**IL-12/23**			
**Ustekinumab**			
	Montero-Vilchez et al., 2022 [158]; Valenzuela-Ubiña et al., 2020 [159]	**Case series (n = 20)** Ustekinumab 90 mg s.c. every 2 months	Successful response to ustekinumab
	Blok et al., 2016 [115]	**Phase II open-label study (n = 17)** Ustekinumab 45 mg s.c. if <90 kg and 90 mg s.c. if >90 kg at weeks 0, 4, 16 and 28	47% achieved an HiSCR at week 40
	Romaní et al., 2020 [160]	**Retrospective multicenter study (n = 14)** Ustekinumab i.v. weight-adjusted induction dose (≤55 kg, 260 mg; 56–85 kg, 390 mg; ≥86 kg, 520 mg) then 90 mg s.c. every 8 weeks	50% achieved an HiSCR at week 16
	Sánchez-Martínez et al., 2020 [161]	**Retrospective unicenter study (n = 6)** Ustekinumab i.v. weight-adjusted induction dose (≤55 kg, 260 mg; 56–85 kg, 390 mg; ≥86 kg, 520 mg), then 90 mg s.c. every 8 weeks	50% achieved an HiSCR at week 12
**IL-23**			
**Guselkumab**			
	Kearney et al., 2020 [163]; Kovacs et al., 2019 [164]; Casseres et al., 2019 [165]; Berman et al., 2021 [166]; Jørgensen et al., 2020 [167]	**Case reports and case series (n = 14)** Guselkumab 100 mg s.c. at week 0 and week 4 then every 8 weeks	Successful response to Guselkumab
	Melgosa Ramos et al., 2022 [168]	**Retrospective bicentric study (n = 11)** Guselkumab 100 mg s.c. at week 0 and week 4 then every 8 weeks; then, for 6 patients, 100 mg every 6 weeks to maintain an HiSCR	63.6% achieved an HiSCR at week 16
	Montero-Vilchez et al., 2020 [169]	**Case series (n = 4)** Guselkumab 100 mg s.c. at week 0 and then every 4 weeks	50% had moderate reduction in IHS4, VAS for pain and DLQI
	NCT0368924 [170]	**Phase II placebo-controlled, double-blind study** Guselkumab 200 mg at weeks 0, 4, 8 and 12 Guselkumab 1200 mg i.v. at weeks 0, 4 and 8, then 200 mg s.c. starting from week 12 Placebo	50% achieved an HiSCR at week 16 45% achieved an HiSCR at week 16 38% achieved an HiSCR at week 16
**Risankizumab**			
	Marques et al., 2021 [172]; Caposiena et al., 2021 [173]; Licata et al., 2021 [174]	**Case reports (n = 4)** Risankizumab 150 mg s.c. at weeks 0 and 4, followed by 150 mg s.c. every 12 weeks	Successful response to Risankizumab
	Repetto et al., 2022 [175]	**Case series (n = 6)** Risankizumab 150 mg s.c. at weeks 0 and 4, followed by 150 mg s.c. every 12 weeks	50% achieved an HiSCR at month 3 and 100% achieved an HiSCR at month 6
	NCT03926169 [176]	**Phase II placebo-controlled study**	Still ongoing
**Tildrakizumab**			
	Kok et al., 2020 [178]; Kok et al., 2021 [179]	**Case series (n = 9)** Tildrakizumab 100 mg s.c. at weeks 0 and 4 and then 200 mg every 4 weeks	100% achieved an HiSCR Reduction in mean AN count of 23.50 from baseline observed at month 15
**IL-1**			
**Anakinra**			
	Tzanetakou et al., 2016 [193]	**Double-blind, randomized, placebo-controlled prospective clinical trial (n = 20)** Anakinra 100 mg s.c. daily for 12 weeks (n = 10) Placebo s.c. (n = 10)	78% achieved an HiSCR at week 12 30% achieved an HiSCR at week 12 After a 12-week observation period, the HiSCR difference between the groups was not significant
	Leslie et al., 2014 [194]	**Open-label phase II study (n = 6)** Anakinra 100 mg s.c. daily for 8 weeks	Mean decrease of 34.8 points in modified Sartorius score. All patients experienced a rebound during an 8-week follow-up off-therapy period.
	Zarchi et al., 2013 [195]; van der Zee et al., 2013 [105]; Russo et al., 2016 [196]; Menis et al., 2015 [197]	**Case reports (n= 5)** Anakinra 100 mg s.c. daily	1 successful response to anakinra 4 failures in response to anakinra
**Bermekimab**			
	Kanni et al., 2018 [199]	**Phase II placebo-controlled, double-blind RCT (NCT02643654) (n = 10)** Bermekimab 7.5 mg/kg i.v. every other week Placebo i.v.	60% achieved an HiSCR at week 12 10% achieved an HiSCR at week 12
	Kanni et al., 2021 [200]	**Open-label extension of NCT02643654 (n = 8)** Bermekimab 7.5 mg/kg i.v. every other week	75% achieved an HiSCR at week 12
	Gottlieb et al., 2020 [201]	**Phase II multicenter open label study (n = 42)** Bermekimab 400 mg s.c. weekly	61% of anti-TNF naïve patients achieved an HiSCR at week 12 63% of anti-TNF failure patients achieved an HiSCR at week 12
	NCT04988308 [202]	**Phase IIa/IIb, multicenter, randomized, placebo- and active comparator-controlled, double-blind, dose-ranging study**	Still ongoing
**Canakinumab**			
	Houriet et al., 2017 [204]; Jaeger et al., 2013 [205]; Tekin et al., 2017 [206]; Sun et al., 2017 [207]	**Case reports (n = 6)** Canakinumab 150 mg s.c. at day 1, then monthly Canakinumab 150 mg s.c. at day 1, day 15, then monthly Canakinumab 150 mg every week/4 weeks/8 week	3 successful responses to canakinumab 3 failures in response to canakinumab
**IL-36**			
**Spesolimab**			
	NCT04762277 [218]	**Phase II placebo-controlled, double-blind RCT**	Still ongoing
**Imsidolimab (ANB019)**			
	NCT04856930 [219]	**Phase II placebo-controlled, double-blind RCT**	Still ongoing

### 3.23. IL6

IL-6 is a proinflammatory cytokine that promotes B-cell antibody production and is involved in multiple inflammatory conditions [220]. Elevated levels of IL-6 and its receptor have been detected in HS skin [80,152], especially in Hurley stage II-III patients compared to healthy controls [78,221].

### 3.24. IL22

IL22 constitutes a member of the IL10 family. It is produced by various types of lymphoid and nonlymphoid cells and is regulated by an interaction of cytokines and transcription factors [222]. IL22 binds directly to IL22 receptors broadly expressed in skin cells and in digestive and respiratory system cells. IL-22 contributes to the preservation of homeostasis against intestinal micro-organisms and commensal bacteria in the boundary organs and tissues [223]. IL22 also stimulates AMPs in epithelial cells and thus prevents bacterial infections and influences inflammatory processes [224]. The upregulation of IL22 has been reported in psoriasis [225]. The ability of IL22 to induce adenoside monophosphates and strongly stimulate the proliferation of keratinocytes agrees with the clinical presentation of psoriasis. IL-17 and IL22 synergistically induce high concentrations of adenoside monophosphates and keratinocyte proliferation, which leads to the thick, dry and scaly epidermis with very few and transient infections [226,227,228,229]. Few data describe the role of IL22 signaling in HS. It has emerged that in HS patients, there is a relative deficiency of IL22 expression in the lesional skin compared with other chronic inflammatory skin diseases, such as psoriasis or atopic dermatitis [151].

Moreover, Jones et al. reported that HS keratinocytes exhibited a significant lower level of IL-1α and IL-22 as well as VEGF compared to normal keratinocytes using an in vitro scratch assay, which suggests that changes in VEGF signaling may be associated with HS pathogenesis [230].

### 3.25. INF

Studies regarding the relevance of IFN-γ are inconsistent [81,231]. A pilot study of HS patients and age-matched chronic wound patients demonstrated significantly elevated IFN-γ levels in the HS effluent patients compared to those with chronic wounds [231]. An increased expression of IFN-γ has also been found in HS lesional skin compared to healthy controls [232]. An analysis of T cells from HS patient skin and blood, however, found no difference in IFN-γ production between HS patients and healthy controls [81].

## 4. Conclusions

Although our understanding of the pathogenetic pathways that drive HS is rapidly emerging, we are just at the beginning. Studies focusing on the characterization of the microbiome, proteome and transcriptome are opening new areas of investigation. However, as mentioned by Zouboulis et al., HS may still be considered a “messy” immunologic disease. Several cytokines have been identified in HS skin samples, but the sequence of the pattern and the key initiators are still to be elucidated. Advances in understanding HS pathogenesis are leading to the expansion of the therapeutic armamentarium. The use of biologics and small molecules accelerates our understanding as they present the opportunity to use clinical effectiveness to validate relevant patterns. The inhibition of TNF-α, IL-1 and IL-17 has been validated in clinical trials as relevant. Recent data suggest that interferon-γ, IL-6, IL-23, C5a and Janus kinase may be successful targets. Although the number of effective agents currently under study is enormously encouraging, future treatment should be guided by personalized therapy, biomarkers and pharmacogenomics, which should drive the therapeutic choice. HS remains a challenging disorder to treat, and patients often need a multidisciplinary approach. Future translational studies are needed to better define this complex skin disease.

## Figures and Tables

**Figure 1 ijms-24-08428-f001:**
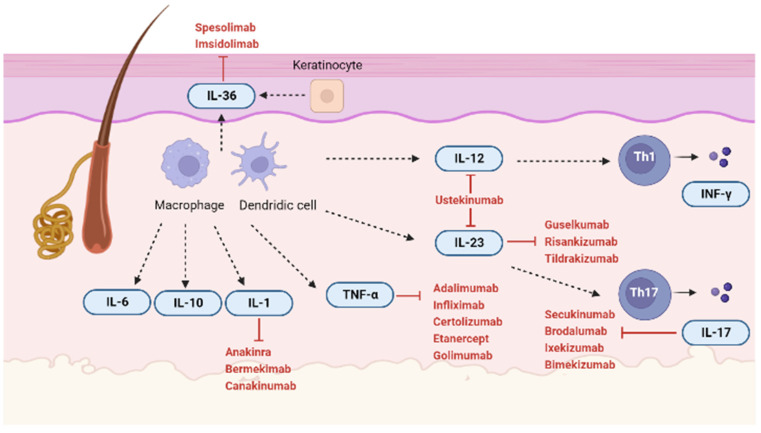
Molecular targets in HS. IL-36: Interleukin (IL) 36; IL-6: Interleukin (IL) 6; IL-10: Interleukin-(IL)10; IL-1: Interleukin (IL) 1; IL-23: Interleukin (IL) 23; IL-12: Interleukin (IL) 12; IL-17: Interleukin (IL) 17; TNFα: Tumour Necrosis Factor α; IFN-γ: Interferon γ (Created by Biorender.com, 21 January 2023).

## Data Availability

Not applicable.

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
