# Peer review of "New Insight into the Molecular Pathomechanism and Immunomodulatory Treatments of Hidradenitis Suppurativa"

_ijms, 2023, doi:10.3390/ijms24098428_

Round 1

Reviewer 1 Report

Thank you very much for opportunity to read manuscript entitled : “New insight into the molecular pathomechanism and immunomodulatory treatments of hidradenitis suppurativa” by Molinelli et al. 

This manuscript is very good. It is a comprehensive presentation on the biological treatment of HS in a way that is accessible to the reader.  All aspects of biological treatment and new drugs are covered here. The paper seems to be accessible to a wide range of readers, not only dermatologists, but also surgeons or doctors of other specialities. 

A great asset of this paper is the broad coverage of the literature (217 references).

I have only 2 comments: please consider extending the section on surgical treatment, although the paper is not about surgery but rather about dermatological treatment, but some aspects of modern surgical techniques should be included to make this manuscript even better.

Introduction section : 

I suggest expanding the section on Quality Of Life - about the effectiveness of surgical therapy on patients' quality of life - PMID: 35893421

I would also suggest, about enhancing this excellent manuscript with the latest methods of reconstructive surgery in the treatment of HS (e.g. co-graft of ADM and STSG) PMID: 36004913 

In my opinion, very important is the non-effective treatment with antibiotic therapy, which according to studies causes the formation of drug-resistant strains in colonised HS lesions. I also suggest this is emphasised in this work to generally emphasise even more the importance of biological treatment. 10.5114/ada.2022.119008

Yours sincerely

Reviewer 2 Report

The authors present an exhaustive review on pathogenesis and new treatments for HS. Some suggestions to improve your paper:

- in the introduction you mention the role of ultrasound in the diagnosis (here you can find the diagnostic criteria for HS and the first sonographic staging system: Wortsman X, et al. Ultrasound in-depth characterization and staging of hidradenitis suppurativa. Dermatol Surg. 2013 Dec;39(12):1835-42.)

- in the same paragraph you mention the possibility to evaluate treatment response by means of ultrasound. In particular, Color Doppler is now considered a valid imaging biomarker to evaluate response during treatment (Nazzaro G, et al. Vascularization and fibrosis are important ultrasonographic tools for assessing response to adalimumab in hidradenitis suppurativa: Prospective study of 32 patients. Dermatol Ther. 2021 Jan;34(1):e14706 - Grand D, et al. Doppler ultrasound-based noninvasive biomarkers in hidradenitis suppurativa: evaluation of analytical and clinical validity. Br J Dermatol. 2021 Apr;184(4):688-696.)

- in the introduction, among the treatment options, you should add laser epilation in mild forms of HS (Nazzaro G, et al. High-frequency ultrasound in hidradenitis suppurativa as rationale for permanent hair laser removal. Skin Res Technol. 2019 Jul;25(4):587-588). I saw that you cited the paper by Molinelli et al but did not comment in the text.

- about pathogenesis, a controversial role of bacterial superinfection should be outlined (Benzecry V, et al. Hidradenitis suppurativa/acne inversa: a prospective bacteriological study and review of the literature. G Ital Dermatol Venereol. 2020 Aug;155(4):459-463 - Corazza M, et al. Irrelevance of Panton-Valentine leukocidin in hidradenitis suppurativa: results from a pilot, observational study. Eur J Clin Microbiol Infect Dis. 2021 Jan;40(1):77-83. - Bettoli V, et al. Rates of antibiotic resistance/sensitivity in bacterial cultures of hidradenitis suppurativa patients. J Eur Acad Dermatol Venereol. 2019 May;33(5):930-936 - Vaienti S, et al. Lymph Node Involvement in Axillary Hidradenitis Suppurativa: A Clinical, Ultrasonographic and Bacteriological Study Conducted during Radical Surgery. J Clin Med. 2021 Apr 1;10(7):1433.)

Thank you for the priviledge of reviewing you paper

Author Response

  • in the introduction you mention the role of ultrasound in the diagnosis (here you can find the diagnostic criteria for HS and the first sonographic staging system: Wortsman X, et al. Ultrasound in-depth characterization and staging of hidradenitis suppurativa. Dermatol Surg. 2013 Dec;39(12):1835-42.): DONE
  • in the same paragraph you mention the possibility to evaluate treatment response by means of ultrasound. In particular, Color Doppler is now considered a valid imaging biomarker to evaluate response during treatment (Nazzaro G, et al. Vascularization and fibrosis are important ultrasonographic tools for assessing response to adalimumab in hidradenitis suppurativa: Prospective study of 32 patients. Dermatol Ther. 2021 Jan;34(1):e14706 - Grand D, et al. Doppler ultrasound-based noninvasive biomarkers in hidradenitis suppurativa: evaluation of analytical and clinical validity. Br J Dermatol. 2021 Apr;184(4):688-696.): We have modified the text based on the reviewer observation.
  • in the introduction, among the treatment options, you should add laser epilation in mild forms of HS (Nazzaro G, et al. High-frequency ultrasound in hidradenitis suppurativa as rationale for permanent hair laser removal. Skin Res Technol. 2019 Jul;25(4):587-588). I saw that you cited the paper by Molinelli et al but did not comment in the text: DONE
  • about pathogenesis, a controversial role of bacterial superinfection should be outlined (Benzecry V, et al. Hidradenitis suppurativa/acne inversa: a prospective bacteriological study and review of the literature. G Ital Dermatol Venereol. 2020 Aug;155(4):459-463 - Corazza M, et al. Irrelevance of Panton-Valentine leukocidin in hidradenitis suppurativa: results from a pilot, observational study. Eur J Clin Microbiol Infect Dis. 2021 Jan;40(1):77-83. - Bettoli V, et al. Rates of antibiotic resistance/sensitivity in bacterial cultures of hidradenitis suppurativa patients. J Eur Acad Dermatol Venereol. 2019 May;33(5):930-936 - Vaienti S, et al. Lymph Node Involvement in Axillary Hidradenitis Suppurativa: A Clinical, Ultrasonographic and Bacteriological Study Conducted during Radical Surgery. J Clin Med. 2021 Apr 1;10(7):1433.): We have modified the text based on the reviewer observation.

Round 2

Reviewer 1 Report

Accept in present form

Author Response

none